# Modulation of Wind-Wave Breaking by Long Surface Waves

Vladimir A. Dulov [1,*], Aleksandr E. Korinenko [1], Vladimir N. Kudryavtsev [1,2] and Vladimir V. Malinovsky [1]

[1] Marine Hydrophysical Institute of Russian Academy of Sciences, 2 Kapitanskaya, 299011 Sevastopol, Russia; korinenko.alex@mhi-ras.ru (A.E.K.); kudr@rshu.ru (V.N.K.); vladimir.malinovsky@mhi-ras.ru (V.V.M.)

[2] Satellite Oceanography Laboratory, Russian State Hydrometeorological University, 98 Malookhtinskiy, 195196 Saint-Petersburg, Russia

[*] Correspondence: dulov@mhi-ras.ru or dulov1952@gmail.com

**Abstract:** This paper reports the results of field measurements of wave breaking modulations by dominant surface waves, taken from the Black Sea research platform at wind speeds ranging from 10 to 20 m/s. Wave breaking events were detected by video recordings of the sea surface synchronized and collocated with the wave gauge measurements. As observed, the main contribution to the fraction of the sea surface covered by whitecaps comes from the breaking of short gravity waves, with phase velocities exceeding 1.25 m/s. Averaging of the wave breaking over the same phases of the dominant long surface waves (LWs, with wavelengths in the range from 32 to 69 m) revealed strong modulation of whitecaps. Wave breaking occurs mainly on the crests of LWs and disappears in their troughs. Data analysis in terms of the modulation transfer function (MTF) shows that the magnitude of the MTF is about 20, it is weakly wind-dependent, and the maximum of whitecapping is windward-shifted from the LW-crest by 15 deg. A simple model of whitecaps modulations by the long waves is suggested. This model is in quantitative agreement with the measurements and correctly reproduces the modulations' magnitude, phase, and non-sinusoidal shape.

**Keywords:** ocean; wind-driven waves; wave breaking; field measurements; whitecap coverage; modulation transfer function

## 1. Introduction

Wave breaking plays a crucial role in various air–sea interactions and remote sensing areas and thus has been the subject of intensive research over the past few decades (see, e.g., [1–5]). Wave breaking of wind-driven waves contributes substantially to air-sea gas exchange [6–8], wave energy and momentum dissipation [9–11], and generation of turbulence in the ocean near-surface layer [12–14]. Wind-wave modeling and forecasting need spectral parameterization for wave breaking [15,16]. Wave breaking affects radar backscattering [17–20] and microwave emission [21,22] of the sea. Wave breaking is also very sensitive to the energy disturbances caused by wave interactions, with sub- and mesoscale surface current gradients being an important component of the wave energy balance. As a consequence, various ocean phenomena, such as internal waves, eddies, current fronts, shallow water bathymetry are displayed on the ocean surface in the form of spatial anomalies of wave breaking parameters tracing the surface current features [23–27]. This opens a promising opportunity for monitoring the ocean dynamics using passive and active microwave satellite remote sensing [28–30]. Due to whitecapping, video processing is a traditional approach for the in-situ investigation of wave breaking statistics and space-time characteristics of "individual" breaking events (see, e.g., [31–34]).

Long surface waves modulate the breaking of short wind waves [35–37], resulting in enhancement of wave breaking around the crests and damping in the long wave throughs. This effect, observed in both the laboratory and the field, has important applications. Modulations of wave breaking lead to variations of radar returns along the wavelength of long waves. This, on the one hand, significantly contributes to the radar modulation transfer



function, which is necessary to retrieve wave parameters from the radar signal [19]. On the other hand, modulations of radar backscattering by long surface waves provide a significant, if not dominant, contribution to the mean Doppler shift of radar backscatter from the ocean surface. Errors of the quantitative estimates of such radar modulations predetermine the accuracy of the ocean surface current retrieval from the Doppler shift anomalies [38–40]. In the context of the air–sea interaction, the effect of wave breaking [35,37] and short-wave spectrum [41] modulations on the aerodynamic roughness along the dominant surface waves profile, significantly amplify (by factor two to three) momentum and energy transfer from the wind to waves [42]. Hence, a better understanding of the wave breaking modulations mechanism can directly contribute to a better understanding of the momentum and energy exchange between atmosphere and ocean on regional and global scales.

Modulations of a certain characteristic of the sea surface, for example, whitecaps coverage, $Q$, by long/dominant surface waves, can be described in terms of the modulation transfer function (MTF), which assumes that $Q$ varies above the long-wave as:

$$Q = \overline{Q}(1 + M\varepsilon \sin(\Phi - \Phi_0)) \tag{1}$$

where $M$ is the MTF magnitude, $\overline{Q}$ means mean value of $Q$, $\varepsilon = KA$ is the steepness of modulating wave with elevations $\zeta(x,t) = A\sin(\Phi)$, $A$ is the wave amplitude, $\Phi = Kx - \Omega t$ is the phase, $K$ is the long-wave wavenumber, $\Phi_0$ is the phase shift between $\zeta$ and $Q$, a positive value of $\Phi_0$ means a lag of $Q$ modulations relative to $\zeta$ (shift of the modulations towards the backward long-wave slope). MTF is the common terminology for describing interactions of short and long surface waves [43–45], radar scattering [19,46], and wave breaking modulations by long surface waves [35,37]. Experimental estimates of whitecaps MTF remain very scarce, and to our knowledge, they are limited by the results reported in [35,37]. According to these data, the magnitudes of the whitecaps MTF are surprisingly large, about 20. This means that wave breaking variations scaled by the mean value are 20 times larger than the steepness of modulating waves, pointing to the nonlinear nature of breaking modulations. Since experimental estimates of wave breaking modulations are extremely limited, new measurements are in great demand to get a deeper insight into the physics of this phenomenon.

In this paper, we report field investigations of the whitecaps modulations by long surface waves performed from a Black Sea research platform, using video recordings of the sea surface. A description of the field experiment, data, and method of data processing is presented in Section 2. Results and interpretation of the data in terms of the MTF revealing large modulations are presented in Section 3. In Section 4, we suggest a simple model of whitecaps modulations that describe quantitatively empirical results. Discussion and conclusion are given in Sections 5 and 6.

## 2. Field Experiment

### 2.1. General Description

The experimental studies were carried out in October 2018 from a Black Sea research platform near the village of Katsiveli. The platform is located about 0.5 km offshore in about 30 m deep water. A more detailed description of the platform and types of wind-wave conditions at this site were reported in [47]. This study was performed under conditions of developing wind waves coming from the open sea without swell.

The wave breaking events were recorded using a digital video camera with a recording rate of 50 frames per second and a resolution of 1920 × 1080 pixels. The camera was mounted at the height of 12 m and directed at 35–40° to the horizontal, which provides a resolution of 2 cm at the water surface. The camera looked opposite to the general direction of dominant waves, which always coincided with the wind direction. Eight video records of length from 60 to 90 min collected at steady wind conditions are used in this study.

Elevations of the sea surface were recorded by a wire wave gauge located in the field of view of the video camera (see Figure 1) at the end of an 11 m long boom to minimize the platform disturbances. The wind speed and direction at a height of 23 m, air temperature

and humidity at a height of 19 m, and temperature in the upper meter of water were measured continuously from the platform using Davis 6152EU meteorological station.

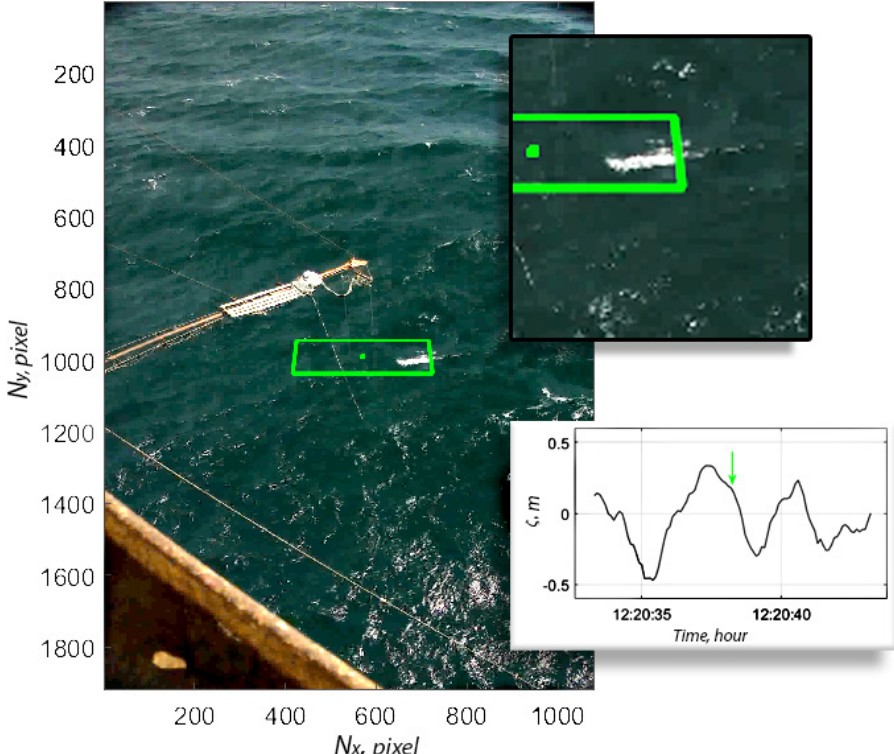

**Figure 1.** Video frame with wave gauge boom and rectangle (in green) of the sea surface for studying the wave breaking modulations. A green point in the rectangle shows the interception of the wave gauge wire with the sea surface. The insets show (**top**) an enlarged fragment of the frame and (**bottom**) a period of the sea surface elevation recording, where an arrow marks the instant of time corresponding to this frame.

Frequency spectra of sea surface elevations, $S(f)$, spectral peak frequencies, $f_p$, and significant wave heights, $H_S = 4\sqrt{\sigma^2}$ ($\sigma^2 = \int S(f)df$ is the elevation variance), were calculated conventionally [48]. The wind speed at 10 m, $U$, was calculated according to the algorithm of Fairall et al. [49]. Mean (over records) values of the wind speed, the spectral peak frequencies, the significant wave heights, the measure of wave ages, $\alpha = c_p/U$ ($c_p$ is the phase velocity of the spectral peak waves), air and water temperatures for all the records are listed in Table 1.

**Table 1.** Measurement summary.

| Run # | $U$ (m/s) | $f_p$(Hz) | $H_S$(m) | $\alpha$ | $t_a$(°C) | $t_w$(°C) | $Q$(%) | Number of Individual Waves | $\varepsilon$ |
|---|---|---|---|---|---|---|---|---|---|
| 1 | 13.1 | 0.17 | 1.2 | 0.7 | 20.6 | 19.5 | 0.36 | 799 | 0.047 |
| 2 | 13.7 | 0.16 | 1.2 | 0.7 | 20.3 | 19.6 | 0.26 | 650 | 0.052 |
| 3 | 14.2 | 0.22 | 1.1 | 0.5 | 21.0 | 19.5 | 0.14 | 1037 | 0.050 |
| 4 | 13.1 | 0.17 | 1.2 | 0.7 | 20.5 | 19.5 | 0.11 | 774 | 0.056 |
| 5 | 18.1 | 0.18 | 1.5 | 0.5 | 19.7 | 19.3 | 0.38 | 814 | 0.058 |
| 6 | 16.0 | 0.15 | 1.9 | 0.6 | 19.5 | 19.3 | 0.22 | 761 | 0.064 |
| 7 | 19.0 | 0.15 | 2.0 | 0.5 | 21 | 19.3 | 0.55 | 467 | 0.071 |
| 8 | 13.4 | 0.15 | 1.8 | 0.8 | 21.4 | 19.3 | 0.21 | 1049 | 0.061 |

### 2.2. Data

Video recordings were processed using the algorithm suggested by Mironov and Dulov [32]. The pixel coordinates in video frames were transferred into horizontal coordinates at a zero-height plane, using known observation geometry. Only the active breakings belonging to "phase A" of Monahan and Wolf [50] were extracted from image sequences. The spots of residual foam left by whitecaps were automatically filtered out based on the distinction of their advancing velocity from the whitecaps one (see [32] for details). Areas of all active breaking at each time instant and their mean velocities were obtained to form the resulting datasets for each of the video recordings. Table 1 shows the whitecap coverage percentage, $Q$, calculated from the data.

Figure 2 shows spectral distributions of whitecap coverage over frequencies, $P(f)$ ($Q = \int P(f)df$), together with wave spectra. The $P(f)$ gives the contributions to whitecap coverage from frequency interval $(f, f + df)$. It differs from Phillips' breaking crest distribution reported in experimental studies (see, e.g., [32–34,51]). As suggested by Phillips [2], the velocity of a breaking front $c_b$ is equal to the phase velocity $c$ of the breaking wave generating the whitecap. Therefore, we obtained the $P(f)$ by converting the measured whitecap velocities to frequency as $f = g/(2\pi c_b)$, using deep water dispersion relation for gravity waves, where $g$ is the acceleration of gravity. On the other hand, if we suggest, following [9,52], that whitecaps move slower than the phase velocity, e.g., $c_b = 0.8c$, the spectrum of whitecaps coverage $P(f)$ is to be modified, as shown in Figure 2.

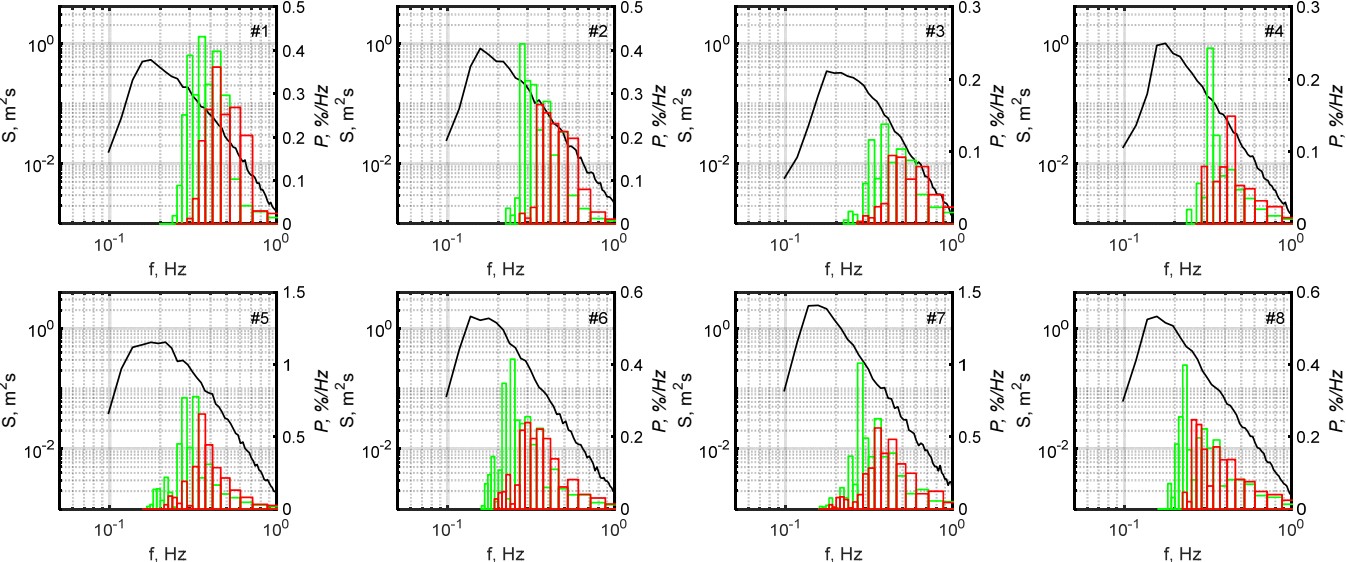

**Figure 2.** Frequency spectra, $S(f)$, (solid line) and distributions of whitecap coverage over frequency, $P(f)$, for $c_b = c$ (red bars) and for $c_b = 0.8c$ (green bars) for all experimental runs.

As follows from Figure 2, only waves from the spectral interval approximately above double spectral peak frequency are breaking. Correspondingly, the length of modulating dominant waves exceeds the length of breaking waves by a factor of three to four. Thus, these data are suitable for analyzing the modulation of wave breaking from the equilibrium range by long waves of spectral peak.

### 2.3. Processing Procedure

The main goal is to compare simultaneously measured whitecap coverage and long-wave elevations that require special synchronization of the video camera and wave gauge data records. To that end, time-series of wave gauge and sound output of the video camera were recorded into a united file. The cross-correlation technique (see, e.g., [53]) was used for

syncing the same time series from wave and video-sound recordings in the data processing. The accuracy of synchronization is 1 millisecond.

Only wave breaking events falling into the rectangular box on the sea surface (see Figure 1) were taken into account for comparison with the wave-gauge signal. The rectangle of $3 \times 1.5$ m$^2$ size was extended along the crests of long waves and centered at the wave-gauge position. Modulations of sea surface brightness visualize the long-wave in video recording (see, e.g., [54]). The coherence of sea brightness and the wave-gauge signal in any point of the rectangle was not less than 0.7, which allows linking whitecaps to the long-wave phase. An error in the determination of the phase can be estimated as:

$$\Delta = \pm 180° \cdot L / \lambda / 2$$

where $\lambda$ is wavelength and $L$ is the rectangle width across the long-wave crest, which for typical wave conditions gives $|\Delta| \leq 5°$.

Figure 3 shows the time series of wave elevations, $\zeta(t)$, and instantaneous whitecap coverage of the rectangle, $q(t)$. Since breaking occurs very rarely, the realization of $q(t)$ has the form of separate spikes against zero background, so the spectral technique for studying the radar MTF, described by Plant [50], is not applicable in this case. Therefore, we used the procedure of individual-wave analysis suggested by Dulov et al. [35]. After smoothing to filter out short waves, the elevation records were split into consecutive periods between zero up-crossings, $\zeta_i(t)$. Augmenting them with corresponding periods of whitecap coverage series, $q_i(t)$, we obtained data subsets $(\zeta_i(t), q_i(t))$ characterizing individual waves (for an example of an individual wave see Figure 3). For each of the individual waves, we introduced the wave period, $T_i$, as the time-length of the subset, and the wavenumber:

$$K_i = (2\pi / T_i)^2 / g$$

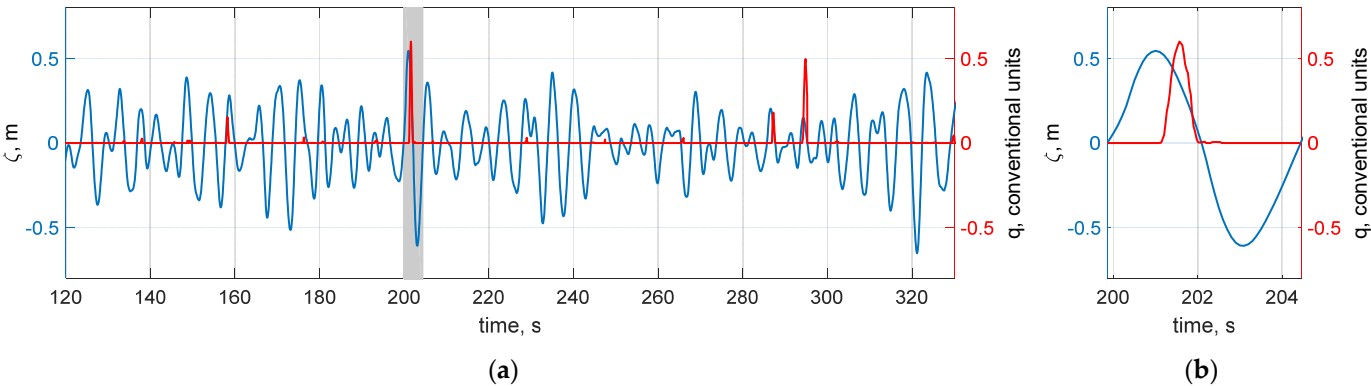

**Figure 3.** An example of time-series (**a**) of wave elevation, $\zeta$ (blue), and whitecap coverage, $q$, (red), and (**b**) zoomed individual wave period corresponding to the grey-marked part of the recording.

To compare different individual waves, we converted the time to the phase of the long-wave:

$$\Phi = 2\pi(t - t_i) / T_i$$

where $t_i$ is the instant of the first zero up-crossing, and considered the dimensionless elevation profiles:

$$\delta_i(\Phi) = K_i \zeta_i(\Phi)$$

Maximum values of $\delta_i(\Phi)$ represent the wave steepness $\varepsilon = AK$ for sinusoidal waves. As a result, several hundred individual-wave subsets $(\delta_i(\Phi), q_i(\Phi))$ were obtained from each of the experimental runs (see Table 1).

## 3. Results

Figure 4 shows averaged individual-wave profiles for each of the runs, $Q(\Phi) = \langle q_i(\Phi) \rangle$ and $K\zeta(\Phi) = \langle \delta_i(\Phi) \rangle$ exhibiting phase-resolved whitecaps distribution. The whitecap coverage profiles were normalized with their mean values (see Table 1). Amplitudes of averaged dimensionless elevation profiles, $\varepsilon$, are also listed in Table 1. Hereinafter, confidence intervals correspond to a double standard error.

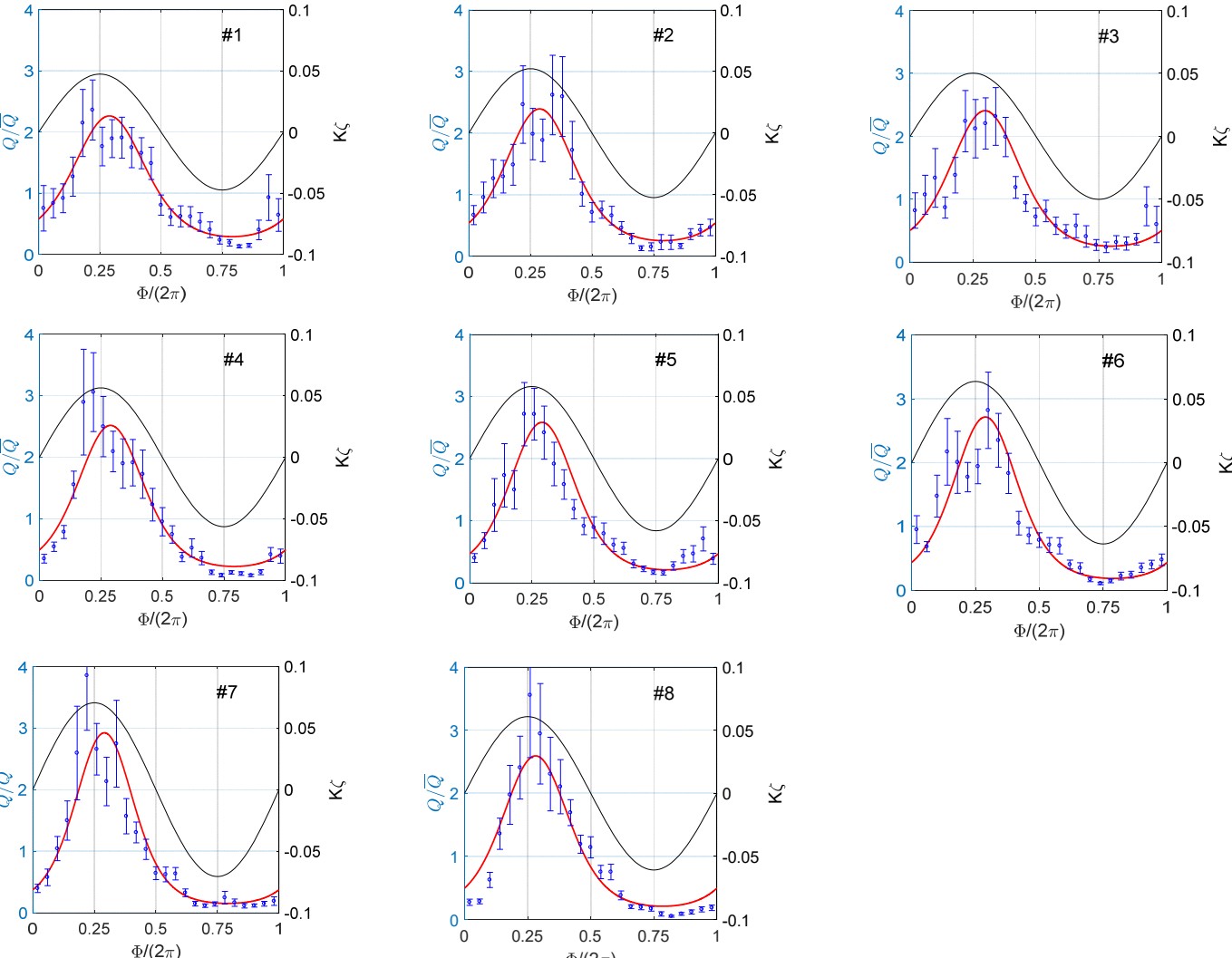

**Figure 4.** Averaged profiles of normalized whitecap coverage $Q/\overline{Q}$ (symbols with double-standard-error bars) and dimensionless wave elevations $K\zeta$ (black lines) along the wavelength of individual waves for all experimental runs. According to the definition of phase $\Phi$, the wave in Figure 4 runs from the right to the left. Red lines show model calculations for measured characteristics $\varepsilon$, $U$, $\alpha$ (see Table 1) and model parameters $n = 5$, $\nu = 0.5$.

In all the graphs, a strong enhancement of wave breaking takes place around the wave crests with a discernible shift to the backward (upwind) slope. Whitecap coverage profiles are clearly deviated from the sinusoidal shape, in contrast to wave elevation, showing a nonlinear connection between them. We describe the wave breaking modulations as:

$$Q = Q_0 \exp(M\varepsilon \sin(\Phi - \Phi_0)) \tag{2}$$

where $Q_0 = \overline{Q} \exp\left(\langle \log(Q/\overline{Q}) \rangle\right)$ and $\langle \dots \rangle$ means averaging over a period of the long wave. For small wave steepness, $\varepsilon \to 0$, Equation (2) reduces to Equation (1), which is a traditional linear description of modulations induced by long-waves in radar signals [46]

or wave breaking [35,37], where $M$ is the modulation transfer function, and positive phase shift $\Phi_0$ corresponds to shift of $Q$ modulations maximum on the backward slope of the modulating wave.

Figure 5a,b, combining all the data, demonstrate better applicability of nonlinear description (2) for data fitting using a sinusoid. Moreover, nonlinear description (2) appears to be inherent in our data, as confirmed in Appendix A. Fitting the $\log(Q/Q_0)$ with $A\sin(\Phi - \Phi_0)$, we obtain that an increase in $\varepsilon$ leads to linear growth of $A$ remaining $\Phi_0$ constant, see Figure 5c,d. Least-square estimates gives:

$$M = 22.9 \pm 2.7° \ \Phi_0 = 14.1° \pm 5.0° \tag{3}$$

which are consistent with linear estimates $M = 23.8 \pm 2.6$ and $\Phi_0 = 6.8° \pm 9.0°$ reported in [35]. It should be noted that the observed phase shift is also in concord with the measurements [36], where a maximum of small-scale breaking detected using IR imagery was found on the rear slope of the modulating waves.

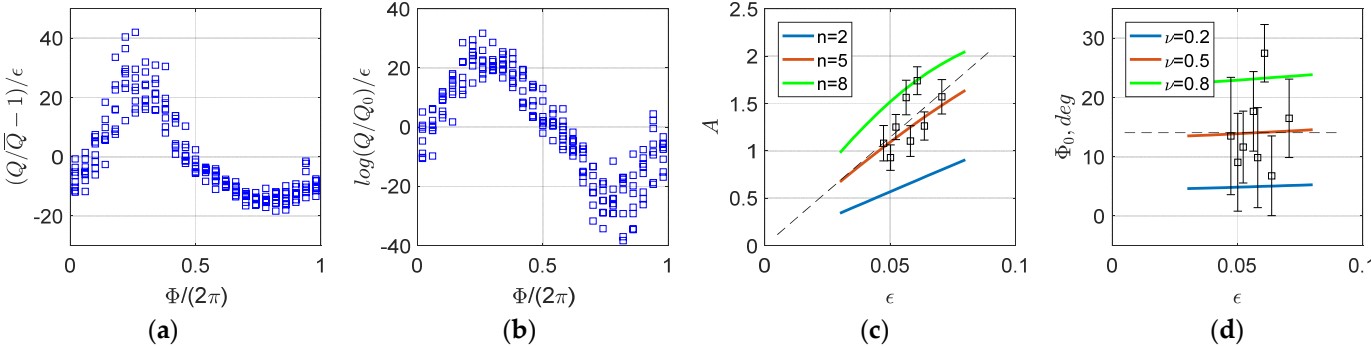

**Figure 5.** To data analysis: collapsing all the data using (**a**) linear (2) and (**b**) nonlinear (1) representations; (**c**) amplitudes and (**d**) phase shifts in fitting the $\log(Q/Q_0)$ using $A\sin(\Phi - \Phi_0)$ according to (1) as a function of long-wave steepness $\varepsilon$ for all recordings. Black dashed lines show least-square estimates (3). Colored solid lines show model calculations for mean data characteristics $U = 14.8 \text{ m/s}$ and $\alpha = 0.63$.

Figure 6 shows wind dependences of modulation parameters. The results of [35,37] obtained using linear representation (1) are also shown for comparison. Though wind dependences are weak, the same trends are evident for all the datasets. Some bias of estimations [37] is probably caused by residual foam, which was not completely removed in their study, and accounting for small-scale, microwave breaking.

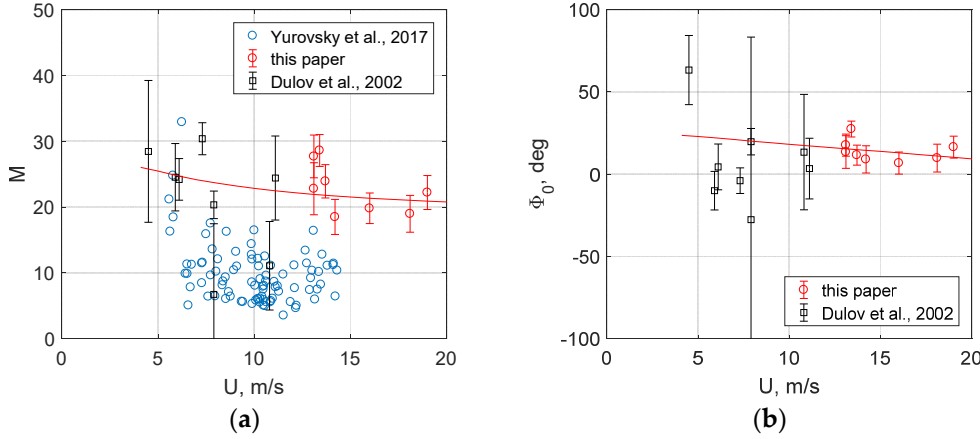

**Figure 6.** Modulation parameters $M$ (**a**) and $\Phi_0$ (**b**) as a function of wind velocity $U$. Symbols show data, lines show model calculations for mean-over-data $\varepsilon = 0.057$, $\alpha = 0.63$ and model parameters $n = 5$ and $\nu = 0.5$.

## 4. Modelling

### 4.1. Governing Equations

Modulations of short wind wave spectrum by long-waves orbital velocities are modeled using the kinetic equation [55,56]:

$$\frac{\partial N(\mathbf{k})}{\partial t} + (c_{gi} + u_i)\frac{\partial N(\mathbf{k})}{\partial x_i} - k_i\frac{\partial u_i}{\partial x_j}\frac{\partial N(\mathbf{k})}{\partial k_j} = S_{in} + S_{nl} - S_{diss} \tag{4}$$

where $\mathbf{k} = (k, \varphi)$ is the short-wave wavenumber vector, $\omega$ is frequency linked to wavenumber via the dispersion relations for deep water, $\omega = \sqrt{gk}$; $N(\mathbf{k})$ is the wave action spectrum, related to the energy (elevation) spectrum as $N(\mathbf{k}) = E(\mathbf{k})/\omega$; $c_{gi}$ is a component of the group velocity of waves, $u$ is the horizontal component of the long-wave orbital velocity, $i$ and $j = 1, 2$. Terms in the right-hand side of (4) describe the wind input, $S_{in}$, nonlinear wave–wave interactions, $S_{nl}$, and dissipation due to wave breaking, $S_{diss}$.

Spectral sources on the right-hand side (4) are not exactly known. Following [2,18,57], we adopt the wind input in the form of:

$$S_{in}(\mathbf{k}) = \beta(\mathbf{k})\omega N(\mathbf{k}) \tag{5}$$

where $\beta(k) = 0.04(u_*/c)^2\cos^2(\phi - \phi_0)$ is the wind growth rate, $u_*$ is the friction velocity, and $\varphi_0$ is the wind direction. Although the nonlinear energy transfer $S_{nl}$ is important for the development of spectral peak waves [15], its role in the equilibrium range dynamics and its modulations is probably not important, and this term is further omitted. Following [2], the dissipation rate can be expressed through the length of wave breaking fronts, $\Lambda(\mathbf{k})$, as:

$$S_{diss}(\mathbf{k}) = b_D\omega^{-1}g^{-2}c_b^5\Lambda(\mathbf{k}) \tag{6}$$

where $b_D$ is Duncan's [1] empirical constant, $\mathbf{c}_b = \mu\mathbf{c}$ is the breaker advancing velocity with $\mu$ varying in the range of $0.8 \div 1$ [2,9,52]. The same quantity, $\Lambda(\mathbf{k})$, defines the fraction of the sea surface covered by whitecaps [2,18]:

$$Q = 2\pi\gamma\int k^{-1}\Lambda(\mathbf{k})d\mathbf{k} \tag{7}$$

where $\gamma$ is the averaged ratio of the whitecap width to the length of wave generating the whitecap, and an integration domain in (7) corresponds to the range of breaking waves observed in an experiment. For the uniform conditions (when waves are stationary and there is no current), the balance between wind input (5) and dissipation (6) results in the following background relationship for $\Lambda(\mathbf{k})$:

$$\Lambda_0(\mathbf{k}) = b_D^{-1}k^{-1}\beta(\mathbf{k})B_0(\mathbf{k}) \tag{8}$$

where $B(\mathbf{k}) = k^4E(\mathbf{k})$ is the saturation spectrum, $B_0(\mathbf{k})$ is its value in the absence of currents (background spectrum). Korinenko et al. [51] tested relationship (8) against the field measurements and found its validity. However, relation (8) is not valid in the presence of the current. On their nature, wave breaking characteristics, e.g., $\Lambda(\mathbf{k})$ as well as the wave breaking dissipation, $S_{diss}$, are strongly nonlinear functions of the spectral level. Following [2,18,54,58], we parameterize spectral distribution of breaking fronts and wave breaking dissipation as power functions of the wave spectrum:

$$\Lambda(\mathbf{k}) = \Lambda_0(\mathbf{k})\left(\frac{B(\mathbf{k})}{B_0(\mathbf{k})}\right)^{n+1} \tag{9}$$

$$S_{diss}(\mathbf{k}) = \beta\omega N(\mathbf{k})\mu^5\left(\frac{B(\mathbf{k})}{B_0(\mathbf{k})}\right)^n \tag{10}$$

where $\Lambda_0(\mathbf{k})$ is defined by (8). These expressions correspond to the Phillips' theory of equilibrium range [2] if $n = 2$ and to the Donelan and Pierson dissipation model [58] if $n = 5$. For the background conditions, when $B(\mathbf{k}) = B_0(\mathbf{k})$, relation (10) is reduced to (6) with (8). As it follows from (9) and (10), any deviations of the wave spectrum, $B(\mathbf{k})$, from the background one, $B_0(\mathbf{k})$, result in an amplified nonlinear response of wave breaking intensity (9) and the energy dissipation (10).

Equation (4) with wind input (5) and dissipation (10) close the model of short-wave spectrum modulation by long surface waves. Once spectrum modulations are found, corresponding modulations of whitecaps coverage could be found from (9) and (7). However, Equations (7) and (9) do not account for the finite lifetime of whitecaps, which is essential when it is comparable with the long-wave period. The reason is the following: whitecap moves with velocity $\mathbf{c}_b$, and during its life span $\tau$, it lags from the long-wave profile moving with velocity $\mathbf{C}$. Since the details of whitecap time-evolution are not exactly known, we introduce this effect phenomenologically, defining $Q$ along LW profile as:

$$Q(x,t) = 2\pi\gamma \int k^{-1}\Lambda(\mathbf{k}, \Phi)d\mathbf{k} = 2\pi\gamma \int k^{-1}\Lambda_0[B(\mathbf{k}, \Phi - \Delta\Phi)/B_0]^{n+1}d\mathbf{k} \quad (11)$$

where $\Phi = \mathbf{K}(\mathbf{C}t - x)$, $\mathbf{K}$, and $\mathbf{C}$ are long-wave phase, wavenumber, and phase velocity, respectively, $\Delta\Phi = \mathbf{K}(\mathbf{C} - \mathbf{c}_b)\tau(k)/2$ is the phase shift between whitecaps and dissipation, $\tau = 2\pi\nu/\omega$ is the whitecap lifetime with $\nu$ as an empirical constant that does not depend on $k$.

We consider the output of the model as:

$$R = Q/\overline{Q} \quad (12)$$

which does not depend on empirical constants $\gamma$ and $b_D$ as follows from model equations. Equations (4)–(11) completely formulate the model of long-wave manifestation in whitecapping. This model corresponds to the models of surface manifestation of the ocean current, suggested in [28,53], augmented with Equation (11) for whitecaps description, which takes into account the finite life span of the breaker.

*4.2. Model Analysis*

We performed model calculations specifying $\mathbf{u}(\mathbf{x}, t)$ in (4) and $B_0$ in (8)–(10) for the case of $\varphi_0 = 0$ as:

$$u_x = \varepsilon\mathbf{C}\sin\Phi \quad u_y = 0 \quad (13)$$

$$B_0 = \begin{cases} \alpha_0\cos^{1/m}\varphi, & \cos\varphi > 0 \\ 0, & \cos\varphi \leq 0 \end{cases} \quad (14)$$

where $m = 2 \div 4$, and $\alpha_0$ is a constant on which the model output (10) does not depend as follows from an analysis of model equations. Such a form of wave spectrum reflects observed wide directional spread of waves at frequencies exceeding twice the spectral peak frequency (see, e.g., [59–61]). To specify the wind growth rate in (5), we calculated the friction velocity $u_*$ according to [49]. Details of model calculations are described in Appendix B. Calculation results very weakly depend on exact values of $\mu$ and $m$. Therefore, further, we present them at $\mu = 1$ and $m = 2$ only.

According to model calculations, the MTF-value, $M$, is mainly determined by parameter $n$, while the phase shift, $\Phi_0$, is mainly determined by parameter $\nu$. Their best matching to data results in:

$$n = 5 \quad \nu = 0.5$$

as illustrated in Figure 5c,d. It is worthy to note that $n = 5$, found here as a value providing the best fit of whitecaps modulations to the data, corresponds to the exponent of the nonlinear dissipation originally suggested by Donelan and Pierson [58] from very different reasoning, to provide the best fit of short-wave spectrum to wind exponent of radar scattering from short gravity waves.

The model reproduces observed whitecap modulations (see Figures 4 and 5) and their wind trend quite well (see Figure 6). Moreover, the model captures finer details of non-sinusoidal whitecap profiles, as demonstrated in Figure A1. These conclusions are related to the conditions of our experiments when the long waves and wind directions were aligned. However, the model simulations shown in Figure 7 suggest that similar features (high values of MTF, weak dependence on wind speed, and the shift of wave breaking to the rear face of the long waves) take place at any angle between the wind and swell. Additional experimental data are needed to confirm these predictions.

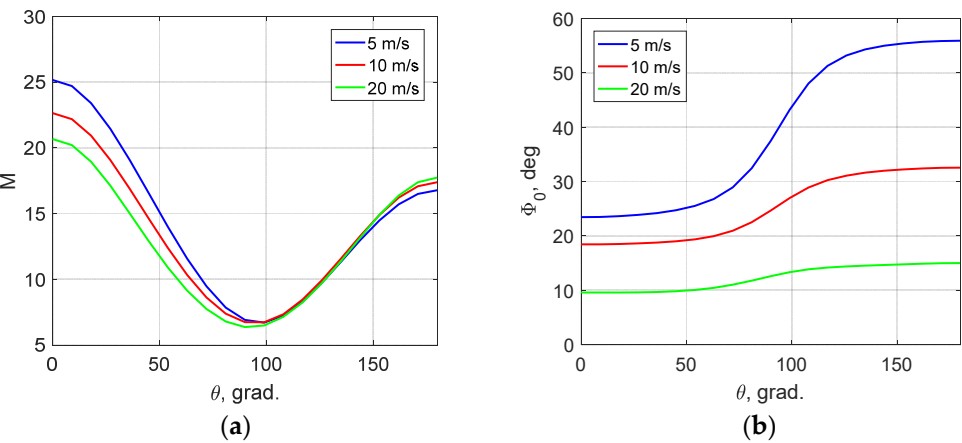

**Figure 7.** (**a**) Magnitudes $M$ and (**b**) phase $\Phi_0$ of the wave breaking MTF as a function of the angle between directions of the wind and long-wave, $\theta$. Legends show wind velocities.

## 5. Discussion

Wave breaking intensity is a strongly nonlinear function of the wave saturation spectrum [2,9]. Therefore, small variations of short-wave spectrum, due to the interaction of waves with the surface current of an arbitrary origin, result in a significantly amplified response of wave breaking to the non-uniform surface currents (see, e.g., [24,25] for internal waves, and [26,28,29] for sub- and meso-scale ocean currents).

In general, as assumed, the effect of long-wave orbital velocities on the short-wave spectrum and wave breaking results from short–long waves interaction, the effect of bound harmonics, and the wind velocity undulations (see, e.g., [36,42,45]). The short–long waves interaction exhibits amplification of short waves on the long-wave crest and their suppression in the long-wave trough at any directions between the wind and the long waves. This fundamental mechanism was studied theoretically [62,63], numerically [64], and experimentally in both the field [43,44] and in the laboratory [45]. Physically, the modulation of short waves caused by short-wave straining and work of the radiation stress against orbital velocities of the long-wave [62]. In spectral representation, this effect in the present study is described by Equation (4).

Wave breaking, namely whitecaps coverage, is a strongly nonlinear function of the wave saturation spectrum. We modeled this effect phenomenologically, introducing a nonlinear link of breaking front lengths with spectral saturation level (9). This nonlinear relationship inevitably leads to a strong response of whitecaps to small wave spectrum modulations, providing high values of the wave breaking MTF $M$. The wave breaking exponent $n$ in (9) is the main parameter defining the value of $M$ (see Figure 5c where slopes of calculated curves are equal to $M$). As whitecaps have a finite lifetime, they lag behind the long-wave crest, providing observed phase shift $\Phi_0$.

This study shows that neither effects of bound harmonics nor air–sea interaction is needed to understand wave breaking modulation by long waves. The short–long waves interaction and nonlinear transfer of short-wave steepness to wave breaking intensity provide a general explanation of wave breaking modulation by long waves, including high values of their modulation transfer function.



## 6. Conclusions

This paper reports results of investigations of modulations of wind-wave breaking by long surface waves using the data taken from the oceanographic platform in the Black Sea and modelling. The data are collected at moderate to high wind conditions, $U_{10}$ is in the range from 10 to 20 m/s, and thus supplement the few field measurements of wave breaking modulation reported before [35,37] for lower wind speeds, $U_{10} < 10$ m/s.

As found, the distribution of whitecaps along the long-wave has a strong nonlinear shape with prevailing whitecapping on the crest of modulating wave and with almost full wave breaking suppression in the trough areas. We proposed a nonlinear representation of the whitecaps modulations in form (2), with a modulation transfer function of about 20. A simple model of whitecaps modulations by long surface waves is suggested, which reproduces the observations on the quantitative level. We modeled the strongly nonlinear response of wave breaking modulations to long waves introducing the wave breaking exponent $n$ (see Equations (9) and (10)). As found, the best fit of modeled whitecaps modulations to the data results in $n = 5$. It should be noted that Donelan and Pierson [58] found the same value for n from another reasoning, in the best fitting of the modeled wave spectrum to wind exponent of radar scattering from short gravity waves.

We anticipate that reported results on wave breaking modulations by long surface waves can be further used in different research applications, in particular for the investigation of radar Doppler scattering from the ocean surface, where hydrodynamic modulations of the radar facets by large-scale surface waves significantly contribute to the formation of the Doppler centroid anomaly [39,40].

**Author Contributions:** Conceptualization, V.N.K. and V.A.D.; methodology, V.A.D., V.V.M. and V.N.K.; software, A.E.K. and V.A.D.; formal analysis, V.V.M.; investigation, A.E.K.; data curation, V.V.M.; writing—original draft preparation, V.A.D. and A.E.K.; writing—review and editing, V.N.K. All authors have read and agreed to the published version of the manuscript.

**Funding:** This work was funded by the Russian Science Foundation Grant No. 21-17-00236.

**Data Availability Statement:** Data is contained within Figures of the article.

**Acknowledgments:** This work was funded by the Russian Science Foundation Grant No. 21-17-00236. Field measurements were performed as a part of the government assignment No. 0555-2021-000421-0004 at Marine Hydrophysical Institute of Russian Academy of Sciences.

**Conflicts of Interest:** The authors declare no conflict of interest.

## Appendix A. On Analysis of the Nonlinear Shape of Whitecap Coverage Profiles

For quantifying the nonlinear connection between profiles of whitecap coverage modulations and longwave elevations, we used truncated Fourier expansion

$$Q = \overline{Q} + Q_1 \sin(\Phi - \Phi_1) + Q_2 \sin(2\Phi - \Phi_2) + \ldots \tag{A1}$$

and described modulations through four parameters:

$$\left( M_1 = \frac{Q_1}{\overline{Q}\varepsilon}, \Phi_1, M_2 = \frac{Q_2}{\overline{Q}\varepsilon^2}, \Phi_2 \right)$$

where $M_2$ and $\Phi_2$ distinguish the nonlinearity. Figure A1a–c shows the dependence of normalized amplitudes and phase shifts of the harmonics on the wave steepness $\varepsilon$. An increase in $\varepsilon$ leads to linear growth of $Q_1/\overline{Q}$ and quadratic growth of $Q_2/\overline{Q}$ remaining the phase shifts constant. Least-square-estimates are:

$$M_1 = 18.3 \pm 1.3 \quad \Phi_1 = 11.6° \pm 3.4° \quad M_2 = 104 \pm 17 \quad \Phi_2 = 107° \pm 12° \tag{A2}$$

For small wave steepness, Equation (2) reads:

$$Q = \overline{Q}\left(1 + \left\langle \log\left(\frac{Q}{\overline{\overline{Q}}}\right)\right\rangle + M\varepsilon \sin(\Phi - \Phi_0) + \frac{M^2\varepsilon^2}{4}\sin(2\Phi - 2\Phi_0 + \frac{\pi}{2}) + \frac{M^2\varepsilon^2}{4} + \cdots\right)$$

that equivalent to (A1) if:

$$\langle \log(Q/\overline{Q})\rangle = -M_1^2\varepsilon^2/4 \tag{A3}$$

$$M_2 = M_1^2/4 \tag{A4}$$

$$\Phi_2 = 2\Phi_0 + \pi/2 \tag{A5}$$

As follows from estimates (A2), Equations (A4) and (A5) are fulfilled in the limits of confidential intervals ($M_1^2/4 = 84 \pm 12 \approx M_2$, $2\Phi_1 + 90° = 113° \pm 7° \approx \Phi_2$). Figure A2d shows the correspondence of data to Equation (A3), where the dashed line is drawn for $M_1^2/4 = 84$. Thus, our data support the representation (2) up to the order of $O(\varepsilon^2)$.

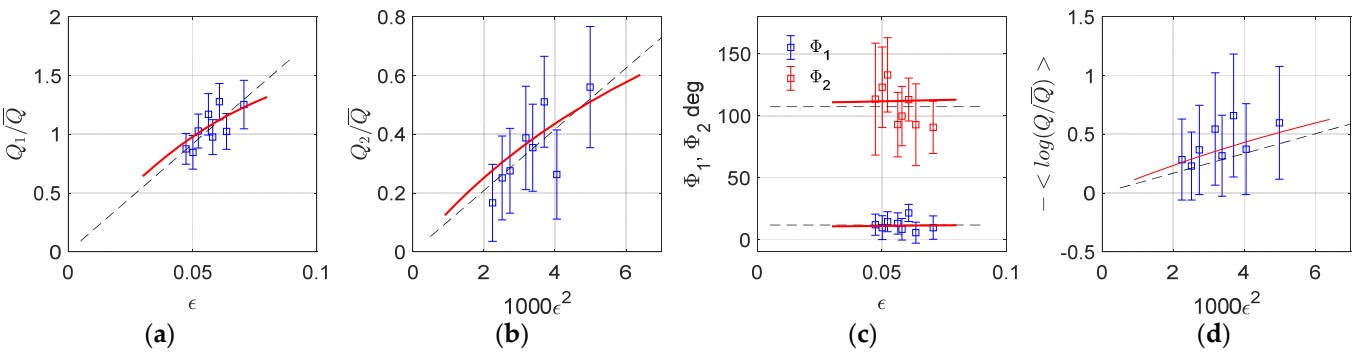

**Figure A1.** Normalized amplitudes (**a**,**b**) and phase shifts (**c**) of the first two harmonics in (A1) as a Figure A2a–c and curve $84\varepsilon^2$ (**d**). Red lines show model calculations for mean data characteristics $U = 14.8$ m/s and $\alpha = 0.63$.

## Appendix B. Calculation Procedure

If the field of currents (13) switches at $t = 0$, then model calculations reduce to solving an initial value problem to equations (4) with initial condition (12) for $B$. The steady-state solutions of the problem at a big enough time model the whitecap response to periodical currents. In Section 4.2, we discussed model results based on this solution.

We considered $B(k, \varphi)$ on the grid log-spaced in $k \in [k_{\min}, k_{\max}]$ and uniform in $\varphi \in [-\pi/2, \pi/2]$, where $k_{\min} = 4K$. $k_{\max} = 6.3$ rad/m that corresponds to the shortest waves, where breaking is resolvable in our experiments. For current velocity (11), after the transition to the frame moving with the long-wave phase speed $C$ through changing variables $(t, x)$ to $(t, \Phi)$, the Equation (4) reads:

$$\frac{\partial B}{\partial \Phi} = F[\Phi, B],\ F = \frac{\varepsilon D \cos\Phi + \beta\kappa B\left(1 - \left(\frac{B}{B_0}\right)^n\right)}{1 - \frac{\cos\phi}{2\kappa} - \varepsilon\sin\Phi} \tag{A6}$$

where $\kappa = \sqrt{k/K}$, and $D = -\cos\varphi\left(\cos\varphi\left(k\frac{\partial B}{\partial k} - \frac{9}{2}B\right) - \sin\varphi\frac{\partial B}{\partial \varphi}\right)$. Numerically, evaluating the term $D$ at every grid node, we considered (A6) as an ordinary differential equation and integrated it with initial conditions at $\Phi = 0$, using a second-order Runge-Kutta scheme.

Figure A2a shows the process of converging to a steady-state solution for the integral measure, fractional whitecap coverage $Q$ normalized with its initial value $Q_0$. After switching on the currents, whitecap production occurs more extensively on long-wave crests than on their troughs due to the nonlinearity of the wave dissipation. It leads to abrupt growth of the mean-over-period value of $Q$, $\overline{Q}$, which is also shown in Figure A2a.

Further whitecap coverage adjusts to periodical disturbances, and $\overline{Q}$ decreases to be slightly lower than its initial value. In this paper, the variable $R = Q/\overline{Q}$ is used both in experiment and model analysis. Though the adjustment process lasts tens of long-wave periods, $R$ remains practically unchanged after the tenth period, see Figure A2b.

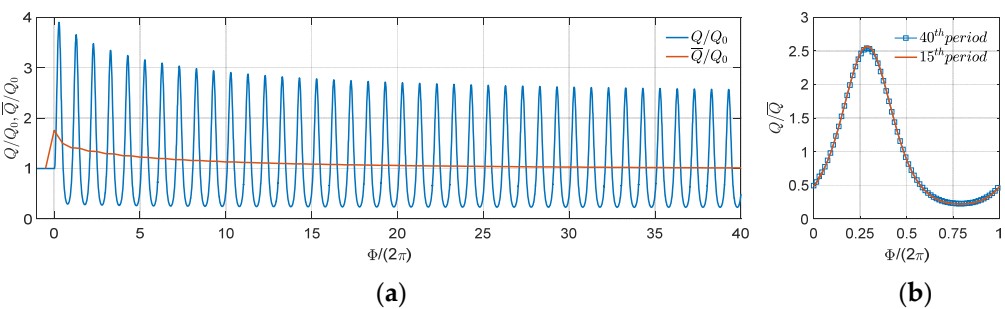

(a)                     (b)

**Figure A2.** Adjustment to periodical currents (11) for (**a**) whitecap coverage $Q$ and its mean-over-period value $\overline{Q}$ and (**b**) their ratio $R = Q/\overline{Q}$ showed for 40th and 15th periods. Calculations were performed for mean data characteristics $\varepsilon = 0.057$, $U = 14.8$ m/s, $\alpha = 0.63$ and model parameters $n = 5$ and $\nu = 0.5$.

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
