# Peer review of "Modulation of Wind-Wave Breaking by Long Surface Waves"

_remotesensing, doi:10.3390/rs13142825_

Round 1
Reviewer 1 Report
The paper focuses on detecting wave breaking events using video recordings and wave gauge measurement. Major revision is needed for further consideration of publication. Detailed comments are:
- Line 27, citation format needs to be improved.
- Line 58, citation format needs to be improved.
- Line 106-107, what does it mean? It should be explained.
- Line 108, row one of Table 1 needs to be revised, such as “U, m/s” should be “U (m/s)”.
- Line 114-115, what is the filtering rule you used?
- Line 148-149, this equation should be introduced in a separate line.
- Line 160-164, each equation in this paragraph should be introduced in a separate line.
- Line 195, reference citation format should be identical throughout the manuscript.
- Line 201, comma at the end of equation should be deleted, it needs to be checked and revised throughout the manuscript.
- Line 251, “Equations (4-11)….” Should be “Equations (4)-(11)….”, the same for line 256.
- Line 258, why m is defined as this value?
- Line 297, it is better to present main conclusions in a separate section.
- Line 319-320, “full stop” should not be put as the starting of the line 320.
- Line 336, does it mean see Equation (7)?
- Line 345, reference format needs to be improved as this journal required.
Author Response
The authors thank the Reviewer for the thoughtful comments on our manuscript. Following review, we made substantial modifications to improve the original manuscript. All of the reviewers’ comments have been taken into account.

Reviewer 2 Report
Modulation of wind-wave breaking by long surface waves
General review:
This manuscript reports on observations of enhanced whitecap activity at the crests of long waves. The enhancement is quantified as a Modulation Transfer Function (MTF), with an approximate value of 20. The work is well-motivated in the introduction, the observations and methods are robust, and the results are strong. However, the second half of the paper is under-developed. The modeling effort is "successful", but the validity is not really shown in figures or discussed in terms of physical process. Furthermore, there are no firm conclusions. This manuscript is an important contribution and should be published with minimal delay, but the model description and conclusions require revision.
Specific points:
The MTF is briefly defined in the introduction (in the paragraph starting at line 56), but its purpose is never fully clarified. This paragraph could be extended to provide more information about various possible applications of the MTF. The authors also mention that a typical value of the MTF found in previous wave breaking studies is approximately 20 - how should one interpret this value in practice?
Table 1 lists the environmental conditions during the field observations. Are all parameters averaged over the same time interval (e.g. 20 minutes)? What is the time interval used for the averaging?
In Figure 2, histograms of breaking activity by frequency are shown with wave spectra (which is a nice representation of the data). In addition to the discussion of whitecap speed relative to phase speed, this would be a good place to mention the literature on the Phillips breaking crest distribution and the literature indicating that breaking crest speed distributions usually have a maximum around half of the peak phase speed.
In section 3, equations (1) and (2) involve the MTF M, which is not defined in the main text apart from the descriptive definition in the introduction. An additional equation could be included after eq. (2), in which the MTF is defined. Also, please fix the equation numbering (equation number (2) appears twice).
In Figure 4, I would call these "phase-resolved ensembles". Regarding the relative phase of whitecaps, it is important to note that all conditions in this study are long waves from the same direction as the short (breaking) waves.
Section 4, the equations are clear, but there is a lack of text describing the processes that cause more breaking at the crests. Do the long waves strain the short waves and modulate steepness? What explanation would be used in a classroom?
Some additional references to consider:
BRANCH AND JESSUP: INFRARED SIGNATURES OF MICROBREAKING WAVE MODULATION, DOI 10.1109/LGRS.2007.895688
Vincent et al, 2019, Prog. in Oceanography, https://doi.org/10.1016/j.pocean.2019.102164
Author Response

(The authors gave the same response as above.)

Round 2
Reviewer 1 Report
I recommend the revised article for publication.
Author Response
The manuscript was kindly edited by Academic Editor Dr. Magdalena Anguelova, and we corrected all the remarks concerning English.
Thank you for your positive response.
